# Patients with Disorders of Consciousness: Are They Nonconscious, Unconscious, or Subconscious? Expanding the Discussion

**DOI:** 10.3390/brainsci13050814

**Published:** 2023-05-17

**Authors:** Andrew A. Fingelkurts, Alexander A. Fingelkurts

**Affiliations:** BM-Science—Brain and Mind Technologies Research Centre, 02601 Espoo, Finland

**Keywords:** disorders of consciousness (DoC), unconsciousness, consciousness, first-person perspective, experiential selfhood, agency, electroencephalogram (EEG), operational architectonics (OA), operational module (OM), vegetative state (VS), unresponsive wakefulness syndrome (UWS), minimally conscious state (MCS), moral status

## Abstract

Unprecedented advancements in the diagnosis and treatment of patients with disorders of consciousness (DoC) have given rise to ethical questions about how to recognize and respect autonomy and a sense of agency of the personhood when those capacities are themselves disordered, as they typically are in patients with DoC. At the intersection of these questions rests the distinction between consciousness and unconsciousness. Indeed, evaluations of consciousness levels and capacity for recovery have a significant impact on decisions regarding whether to discontinue or prolong life-sustaining therapy for DoC patients. However, in the unconsciousness domain, there is the confusing array of terms that are regularly used interchangeably, making it quite challenging to comprehend what unconsciousness is and how it might be empirically grounded. In this opinion paper, we will provide a brief overview of the state of the field of unconsciousness and show how a rapidly evolving electroencephalogram (EEG) neuroimaging technique may offer empirical, theoretical, and practical tools to approach unconsciousness and to improve our ability to distinguish consciousness from *un*consciousness and also *non*consciousness with greater precision, particularly in cases that are borderline (as is typical in patients with DoC). Furthermore, we will provide a clear description of three distant notions of (un)consciousness (*un*consciousness, *non*consciousness, and *sub*consciousness) and discuss how they relate to the experiential selfhood which is essential for comprehending the moral significance of what makes life worth living.

## 1. Introduction

Recent unprecedented advancements in the diagnosis and treatment of a uniquely vulnerable and incapacitated population of patients with disorders of consciousness (DoC) are rapidly raising neuroethical concerns [1,2]. Neuroethics refers to the ethics of neuroscience, that is, what is acceptable and unacceptable in terms of evaluating or manipulating the nervous system in clinical care or research in the neuroscience domain [3,4,5].

Particularly pertinent are the questions of how to detect and respect autonomy and a sense of agency of the personhood when the capacities that constitute autonomy and agency are themselves disordered, as they usually are in patients with DoC [1,6]. Such questions are not a trivial or academic curiosity but rather an important inquiry as some theorists have argued that states of diminished or absent consciousness may preclude attribution of a ‘full moral status’ or the experience of ‘life worth living’ to patients classified as having DoC [2,7,8].

According to the existing clinical nomenclature and diagnostic criteria, DoC refers to a ‘family’ of pathological states that characterize patients who, after a period of coma, have regained the wakefulness–sleep cycle but lack the ability to display overt behaviors [1,9,10]. These states are: (i) vegetative state (VS) [11], which has recently been referred to as unresponsive wakefulness syndrome (UWS) [12] and is described as a ”clinical condition of complete unawareness of the self and the environment” ([13], p. 1499); (ii) the minimally conscious state (MCS) [14], which currently encompasses MCS- and MCS+ [15] and is defined as ”a condition of severely altered consciousness in which minimal but definite behavioral evidence of self or environmental awareness is demonstrated” ([14], pp. 350–351); and (iii) patients who have emerged from an MCS (EMCS and related states) [16] and are in a confusional state (mostly conscious but cognitively impaired) [17,18].

As Young has wrote “…there are a wide range of intriguing philosophical puzzles relevant to this field, relating to the proper classification of border-zone states of consciousness, the relationship of consciousness and personal identity, how to reconcile the subjectivity of consciousness with our conception of an objective reality and the relationship between neural processes and phenomenal experiences” ([1], p. 3293; see also [19,20,21]). At the crossroads of these questions lies the distinction between consciousness and *un*consciousness. Indeed, decisions about whether to limit or continue life-sustaining treatment of DoC patients are heavily influenced by assessments of consciousness presence and its recovery capacities [21,22]. Expectations of poor consciousness recovery and associated attributions of therapeutic futility typically underlie the decision to discontinue life-sustaining treatment [23]. This highlights how important consciousness/unconsciousness boundary is to the idea of personhood/selfhood and to what makes life worth living [1,6]. Increased awareness of these issues and clarity regarding terminology are particularly timely given the recent urgent need to maximize ethically responsible care in this population of patients.

While the view on consciousness as a lived subjective experience that is immediately present to a subject right now (subjective present) and right here (subjective space) [24,25] is currently supported by most researchers working in the field [26,27,28,29,30,31,32,33,34], when it comes to unconsciousness, there is a maze of terms that are frequently used interchangeably, making it very difficult to comprehend what unconsciousness is and how it might be empirically grounded. As Kozyreva rightly points out, “even though the unconscious is no longer a scandal for science and philosophy, it still holds strong positions as one of the most challenging topics for the research of the human mind” ([35], p. 200).

In this opinion paper, we will provide a snapshot of the current state of affairs in the field of unconsciousness (Section 2) and demonstrate how a rapidly developing electroencephalogram (EEG) neuroimaging technique (Section 2.1) may provide empirical, theoretical, and practical tools to approach the unconsciousness and to improve our ability to distinguish consciousness from unconsciousness with greater precision, particularly in cases that are borderline (Section 2.2) or involve diminished experiential selfhood (Section 3). Even though “…these technologies remain imperfect and cannot replace behavioural measures, their ability to detect consciousness missed by behavioural measures challenges longstanding historical and categorical reliance on behavioural measures in the ascription of conscious states, both in clinical practice and in philosophical tradition” ([1], p. 3293).

## 2. (Un)consciousness

When one looks at the many different ways the term ‘(un)consciousness’ has been used in literature, it becomes clear that cognitive, psychological, psychoanalytical, and philosophical literature confounds at least three distinct senses in which a phenomenon could be understood and conceptualized: *un*consciousness, *non*consciousness, and *sub*consciousness. Additionally, there are related though more fuzzy terms such as ‘implicit consciousness’ [36], ‘a state in which it is not like anything to be’ [37], ‘passive level of subjective experience’ [38], ‘objectless awareness’ [39], ‘content-free awareness’ [40], ‘the unperceived in the perceived’ [41], ‘appearance of the non-appearing’ [42], ‘the manifestation of absence’ [43], and ‘the state of nothingness’ [39]. Further, Eastern (Buddhist and Vedanta) philosophers have proposed yet other notions aiming to characterize unconsciousness. The most common Eastern notions are ‘the basic/storehouse mind’, “…which is a subliminal mind or baseline (unmanifest) consciousness that carries along in it seeds of all karmic potentials and latent dispositions…, including forthcoming manifest conscious states that are bound to arise from a series of moments” ([44], p. 130; see also [45,46]), and ‘luminosity’, which is a state distinct from waking and dreaming and lacking any sort of cognition or perception [47,48] (for a more scientifically grounded analysis of these terms, see [49,50,51]).

In the following, we will primarily focus on the three distant concepts of (un)consciousness mentioned above, though we may touch on other meanings as needed for the discussion.

The term ‘*un*consciousness’ was first mentioned in the early 1800s to describe hypnotically induced behavior in which the subjects were unaware of the reasons and causes behind their actions [52]. Similarly, later, Freud, who was well acquainted with early hypnosis research (see [53]), used the term to refer to behavior and ideation that were not consciously caused or intended. Therefore, this conceptualization of the term ‘unconsciousness’, whereas unconscious phenomenon is equated with unintentional, was used within the field of social psychology for roughly a quarter of the past century [54]. Continuing along this line of thought, and employing the newly established paradigms of studying brain activity, a number of researchers [55,56,57,58] postulated that consciousness is not the source or origin of human behavior; instead, impulses to act are first activated unconsciously in the brain, and later they are ‘claimed’ by consciousness and experienced by it [59,60] as the end result of *un*conscious influences on behavior, thought, and action without us realizing it [36,61,62]. It is abundantly clear that the usage of the term ‘unconsciousness’ in this manner applies to a very wide range of phenomena under the presumption that they all possess the same fundamental quality of (un)consciousness. Roughly, these are (a) mental states that lack phenomenal awareness on the one hand and (b) multiple brain’s neurophysiological (*non*-phenomenal or *non*conscious) processes on the other. The latter is also frequently referred to as ‘subliminal perception’ to delineate the influence of events in the current stimulus environment that cannot be consciously perceived but ‘noticed’ (and processed) by the brain. This was studied in such experimental models as blindsight [63,64], masked semantic priming [65], unconscious perception in prosopagnosia [66], and many others [61].

We argue here that, since the terminology used to describe various states of (un)consciousness has implications for clinical practice as well as for fundamental scientific research, there are compelling reasons to critically re-examine our ‘traditional’ use of the (un)consciousness concept in order to improve clarity, diminish overlap, and, in this way, to establish a stronger foundation for subsequent clinical, scientific, philosophical, and neuroethics research. Indeed, the so-called ‘jingle fallacy’, in which we use the same term to refer to different phenomena, and the ‘jangle fallacy’, in which we use different terms to refer to the same phenomenon [67], both get in the way of understanding, preclude the accumulation of fundamental knowledge, and impair progress.

Keeping this in mind and following the work of Searle [68], Velmans [69], Revonsuo [28], McFadden [70], and Hales [71], we are proposing the following, a more nuanced, *tripartite* definition for the (un)conscious (Figure 1) that avoids the trap of lumping together various ‘flavors’ of the phenomenon (a shorter version of these was suggested previously in [24,25]):
(i)*Non*consciousness—it does **not** belong to the mental/experiential/phenomenal domain (Figure 1); it is the myriad of neurophysiological, physical, and biological processes that take place exclusively in the brain (and also in nervous system) *outside* of the ‘mind-space’ [72]. They are always out of reach, i.e., inaccessible for mentality or phenomenal consciousness, and, hence, referring to unconsciousness as part of the brain’s physical (*non*mental) mechanisms makes little conceptual sense [73]. This level of organization can be understood as an autonomous, fast, self-organizing, dynamic system that acquires, processes, stores, and retrieves information to secure its own wellbeing and survival, and the vast majority of life forms ‘possess’ it—though to varying degrees depending on their complexity [74,75,76]. Therefore, crucially, phenomenal consciousness is not necessary for information processing or for adaptation of the organism in general [77]. It is noteworthy that this level encompasses all physiological processes in entirety and is not restricted to any particular localized neural circuit or brain region.(ii)*Un*consciousness—it **belongs** to the mental/phenomenal domain (Figure 1), although it *lacks* phenomenal awareness at any given time and therefore is not accessible for voluntary control (it cannot be inhibited, suspended, or terminated [78]) or for rational expression (subjectivity without awareness [79]). However, it can have an impact on various aspects of phenomenal consciousness, including motivation, feelings, goals, behavior, and decision making [36,54,61]. Because it shares sophisticated characteristics with its conscious counterpart [80], it determines significant portions of our personality, skills, preferences, and experience, and it is responsible for important aspects of our ability to adjust and function effectively [81,82]. At the same time, it is not always integrated with the knowledge and beliefs that are held consciously, and it may even sometimes be inconsistent with them, resulting in severe conflicts and occasionally leading to mental health issues [83].(iii)*Sub*consciousness—it also **does fall** under the mental/phenomenal domain (Figure 1) and refers to a part of the mind that is *not* at any given moment in the focus of attention but which *has the potential* for bursting into consciousness [84,85]. According to Jung, ”Such material has mostly become unconscious because—in a manner of speaking—there is no room for it in the conscious mind. Some of one’s thoughts lose their emotional energy and become subliminal (that is to say, they no longer receive so much of our conscious attention) because they have come to seem uninteresting or irrelevant, or because there is some reason why we wish to push them out of sight. It is, in fact, normal and necessary for us to ‘forget’ in this fashion, in order to make room in our conscious minds for new impressions and ideas” ([86], p. 37). Normally, attention enables the rapid actualization of subconscious information and its availability for conscious experience at any given temporal period [61,87,88] (for an extensive analysis and discussion, see [85]).
Figure 1Tripartite definition for the (un)consciousness. Explanations are in the text.
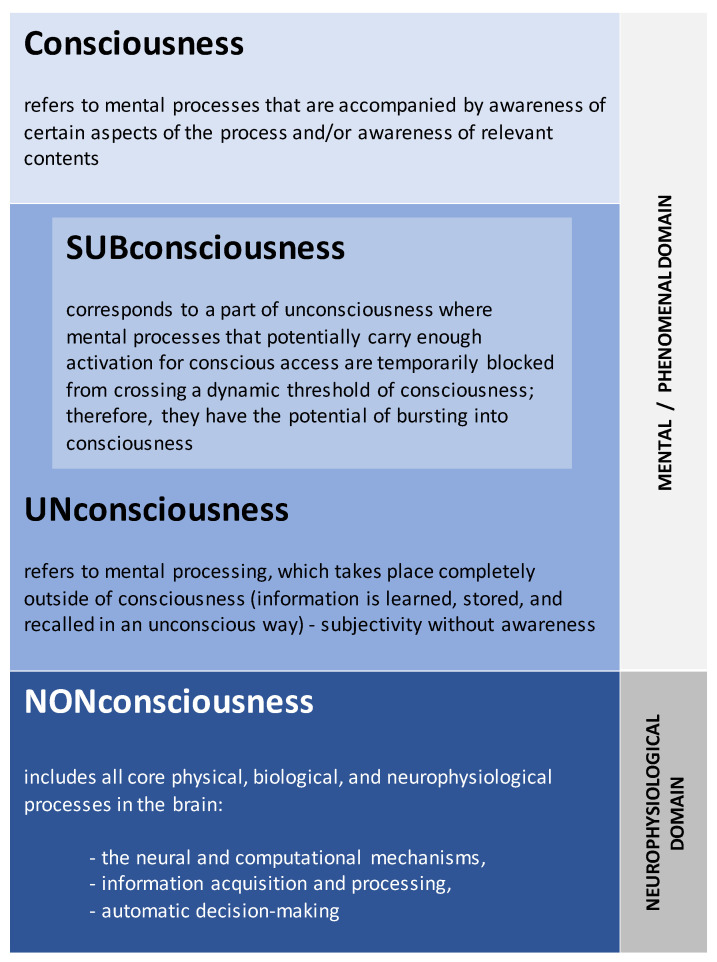



Next, we will advance the analysis of these three phenomena (described by three concepts above; Figure 1) by exploring in more detail their neurophenomenology and mutual relationship within the common conceptual framework (Operational Architectonics [24,25]) stemming from the advanced analysis of the EEG neuroimaging technique [89,90,91]. Operational, in this context, refers to the notion of ‘operation’, which is both a process lasting in time and simultaneously an event that has a beginning and an end [24,25]. Until we are able to dissect such neurophenomenology and accurately describe the various components that comprise it, we cannot expect to adequately connect these phenomena to a functioning human brain. These requirements must be met in order to determine a model that is both neurophenomenologically plausible and practically useful in helping to distinguish consciousness from *un*consciousness, including so-called ‘covert consciousness’ (which is a subjective experience that is present despite of the absence of overt signs of self-expression or deliberate motoric responses) [92]), which is a major challenge in patients with DoC.

### 2.1. Neurophenomenology of Non-, Un-, and Sub-Consciousness

According to the empirically grounded neurophenomenological framework of the Operational Architectonics (OA) of brain and mind functioning [24,25,93] (Figure 2), the mind phenomenological nested architecture and the brain operational nested architectonics are complementary aspects of the same unified metastable continuum [94], which is instantiated by the interaction of two complementary functional tendencies of cooperative integration and autonomous fragmentation [95]. In brief, the OA theory states [24] that the brain ‘constructs’ a continuum of dynamic *electro–physiological* spatiotemporal patterns from a multisensory stream of *neural events* triggered by the spatiotemporal patterns of the outside physical environment (external physical space–time, EPST) (Figure 2). The brain’s internal physical space–time (IPST), via the reordering and recombination of signals from the outside physical environment (EPST), is responsible for such operations [24]. The brain’s IPST level transforms the EPST relations into highly organized and dynamic spatiotemporal patterns of the nested *extracellular electromagnetic fields* generated by local transient and functional neuronal assemblies, which are the source of volumetric, operational spatiotemporal patterns (OST level) [24] (Figure 2). These operational patterns of electromagnetic fields, also known as operational modules or OMs, directly self-present *phenomenal spatiotemporal patterns* at the higher level of abstractness—phenomenal space–time (PST) (see also [71]). The PST, as a whole, in turn serves as a transparent surrogate of the EPST of the world [24]. The term ‘transparency’ refers to the subjectively experienced phenomenal contents being completely transparent, in that they present only what appears to be actual patterns, objects, or scenes existing in the physical world, as opposed to some sort of ‘virtual simulations’ (transparent phenomenal surrogates) of the things that they are presenting [96]. Now that we have this overview in place, we can move on to the topic of our analysis.

The lowest level of the OA brain–mind nested hierarchy is presented by highly distributed and intermixed *cortical neurons and their neuropil* that consists of a dense tangle of axon terminals, dendrites, synaptic connections, and glial cell processes [97,98]. This neuronal net together with its neuropil and related complex physiology constitutes an internal structural analog of 3D space and time (IPST)—some sort of distributed coordinate matrix in the brain—which has no phenomenal functions whatsoever. At this level, only the most fundamental (elemental) physical operations of the brain are carried out by neurons: these functionally autonomous, self-organizing operations process the electric currents which arrive on neuron dendrites, integrate them, and transmit the resulting electrical current to other connected neurons through its axons [24] using so called ‘predictive principle’, which regulates and controls the state of the brain as a whole [99]. Such operations have only a neurophysiological ontology; they are unable to directly alter or change the content of subjective experience (phenomenal consciousness) [25], and, according to Revonsuo [28] and Searle [68], they are totally outside of mental/subjective domain, and therefore they are **entirely *non*conscious**. Indeed, as empirical studies have shown, the activity of neurons does not correlate reliably and predictably with higher cognition and levels of consciousness (for an extensive review and discussion, see [24,100]), even though, as was documented using in vitro neuron cultures (including so-called ‘brain organoids’) [101], they can perform a number of cognitive and memory operations, such as learning, source separation [102,103,104], and even controlling physical robotic systems and simulated video games [105,106]. When it comes to phenomenology, then “…we never experience subjectively the contentless coordinate system as such directly; we could know about it only through the relations among phenomenal objects” ([24], p. 212) that are ‘located’ at another (higher) level of the OA brain–mind nested hierarchy (see below). It is interesting to note that it has been speculated recently that this subphenomenal coordinate matrix may, theoretically, be introspectively available to some extent; however, normally humans never direct their attention there because either culturally they never expect that something phenomenological can actually be discovered there or because of some particular ‘introspective neglect’ (a naturally evolved inability to direct attention there) [51]. Despite this, in rare cases of deep meditation, cultivated by the contemplative Vedic and Buddhistic traditions, and primed by their cultural heritage and teachings [107], it is claimed that a unique state could be reached in which subjects are experiencing an empty void [108]. Often referred as ‘pure consciousness’ by these traditions, it is characterized by an ‘emptying out’ of all experiential content and phenomenological qualities, including concepts, memories, thoughts, sense perceptions, images, and self as well [109]. However, it is necessary to keep in mind that Eastern traditions frequently fail to clearly distinguish between phenomenological, epistemological, and metaphysical readings of concepts such as ‘pure consciousness’ [51]. Therefore, we argue here that, after analytically separating phenomenological descriptions from the metaphysical interpretations in the relevant literature, it is highly likely that the descriptions of contentless states of the quiet mind relate to the next *un*conscious (but already phenomenal) level in the OA of the brain–mind hierarchy, which is opposite to the lowest neuro-physical *no*nconscious (nonphenomenal) level. The only exception might be (and this is just a speculative suggestion) an extremely unique postmortem meditative state cultivated in Tibetan Buddhism and referred to as *thugs dam* (pronounced tukdam) [107]. It is called ‘postmortem’ because, on rare occasions, while being engaged in this particular type of meditation, the accomplished Tibetan Buddhist practitioners pass away and remain in this state for a week or even a month without decomposing or smelling [110]. Externally, after clinically confirmed death, this extraordinary meditative state is manifested “…as a delay in, or attenuation of, the processes of postmortem decomposition. The visage of those in tukdam is described as radiant, their skin remains supple and elastic, and the area around the heart is said to be warmer than the rest of the body” ([111], p. 2). Internally, it is believed that the tukdam state is experienced as a ‘pristine luminous expanse’, ‘unimpeded radiance’, and inseparable from ‘ground emptiness’ [107,112]. We propose here that those descriptions may be an attempt to analogously describe, in human language, certain distributed, self-organized, functional, and dynamic properties of the purely neurophysiological (non-intentional) level as such, which is largely independent from the dynamic kaleidoscope of phenomenological patterns, objects, and states found at the higher levels of OA brain–mind hierarchy [24,25] (for some further relevant discussions, see [51,113,114,115]) and normally is inaccessible for phenomenal awareness. At least in the meditation that may lead to the tukdam state, it was demonstrated with the assistance of EEG analysis that there is a gradual decrease in the brain responses to the stimuli from the external world accompanied by the progressive withdrawal of phenomenal consciousness [116].

The next higher level of the OA brain–mind nested hierarchy is instantiated by the so-called mesoscopic electric activity, which is the multitude of maximally distributed *endogenous local electromagnetic fields* generated by functional and transient neuronal assemblies [24,25] (see also [117,118]), “…where fields emerge from and within other fields, with multiple levels of fields, and fields being mutually constitutive of other fields, to collectively present in the mind a nested and dynamic phenomenal world” ([44], p. 130). Such a spatially organized matrix of local electromagnetic fields, similarly to a previous *non*conscious neurophysiological level, is in constant flux in the form of processes (operations) that carry potentialities and latent dispositions that have the ability to realize any characteristic variety of self-presenting qualitative features [119,120,121]. These phenomenal features (qualities) are the identity, the ‘stuff’ that experiences per se are made of, and they can be described in terms of the simplest phenomenal contents (sounds, colors, touches, emotions, tastes, smells, etc.) [25]. This multidimensional, maximally distributed phenomenal space of possible phenomenal states and features, a space with many different points, regions, or trajectories [122], constitutes the phenomenal (albeit *un*conscious) counterpart of the *non*conscious physical 3D matrix of neuronal nets (see above). It is unconscious because the phenomenology that corresponds to this level is stubbornly evasive and, if made incidentally explicit through deep meditation or psychedelic use [123], its essential phenomenal character cannot be compared to any other familiar form of conscious content we know: it is ubiquitous and all-permeating, implicitly underlying all other, more complex, phenomenal self-presenting forms and capable of modulating or even changing them [51] (for some empirical support, see [124]). In this sense, “…this phenomenal space in which all experiences take place forms a bridge between nonconscious biological mechanisms and phenomenal consciousness” ([24], p. 212; for a further discussion, see also [28]). Again, this level is not restricted to any particular localized neural circuit or brain region—this is the functional property of the whole brain.

The phenomenally conscious world (which is yet next level in the OA hierarchy) has an immensely intricate structure and finely nested hierarchical organization [25,28], whereas the phenomenal patterns of various modalities (such as visual and auditory) are spatially and temporally integrated, allowing for the realization of multiple features belonging to the same phenomenal object in the same location and temporal interval [24]. Phenomenal objects here are defined as complex patterns of phenomenal qualities that are spatially extended and bounded together to form a unified ‘item’ (Gestalt [125]) with a specific meaningful categorization (semantics [126]) immediately present for the subject. Any such phenomenal object can be further organized in a hierarchical fashion to function as a part (or feature) of a more complex phenomenal object, or, alternatively, it can be broken down into its constituent parts, which can all be realized as distinct, simpler phenomenal objects that are independent of one another and possess their own Gestalt and semantic properties [24,25]. This entire dynamic complexity of the phenomenal objects and states is substantiated by the complex dynamics of nested 3D local electromagnetic fields and their more complex and non-local aggregates in the form of OMs (see above) [24,25]. The complexity of phenomenality is linked to the complexity of the electromagnetic fields generated by a set of transient functional neuronal assemblies which are located in different regions of the brain. The exact set varies contingent on the involved qualitative phenomenal features they present (for further discussion, see [24,25]). Importantly, in order to establish spatial and meaningful relations among themselves, all levels of this nested hierarchical architecture of phenomenal consciousness (phenomenal qualities, their patterns, and full-fledged phenomenal objects, scenes, or concepts/thoughts, etc.) are simultaneously co-present [94,127,128].

The phenomenal objects or thoughts that are not actualized at the moment (being pre-attentive or not in the focus of attention) can be described as raw (or candidate) objects and thoughts that do not yet possess full-fledged Gestalt and semantic properties but are rather some phenomenal semi-defined ‘stuff’ [28], thus constituting a *sub*conscious level (with its potential for bursting into consciousness) [84,85]. Normally, it is the attention that directs (either through a self-organized, bottom-up, innate mechanism or by means of a top-down, focused, intentional process) the actualization and sustainability of fully conscious phenomenal objects, thoughts, memories, and decisions, moving serially from one phenomenal object, thought, memory, or decision to another, on a one-at-a-time basis [85,129,130] (for a relation to neurophysiology and concrete brain areas/circuits, see [129]). Crucially, the ability to voluntary direct one’s attention internally (within the mind) is likely exclusively a human skill; it enables the essentially unlimited and temporally extended combination of phenomenal images, symbols, or thoughts, independent of the actual presence of external (environmental) stimuli or specific training [131,132]. The entire process gives rise to a stream of consciousness [133] in which the phenomenal content, which is constantly changing, briefly ‘frizzes’ before abruptly switching to a new one. As Freeman put it [134], the stream of consciousness is ‘cinematographic’ rather than continuous, with multiple frames coalescing into rivulets. To characterize this phenomenon, Metzinger [135] introduced the concept of mental presentation, which is a subjective window of ‘presence’. According to OA, the succession of discrete and relatively stable complex aggregates of electromagnetic fields (in the form of OMs), separated by rapid transitive processes, instantiates the succession of complex phenomenal images, memories, or thoughts, thus presenting a cinematographic stream of conscious experiences [24,25].

In this way, the OA framework of brain–mind functioning yields a plausible and neurophenomenolgically grounded foundation for understanding how phenomenological richness of consciousness emerges from the nested hierarchy of brain architectonics of electromagnetic fields. On a practical note, as it is clear from the above description, it provides a clear picture how the EEG-based OA analysis could aid the inquiry into the evaluation of the presence of phenomenal contents at the boundaries of consciousness, including covert consciousness, as well as separation of the *un*conscious phenomenal states from *non*conscious (purely neurophysiological) ones, which is crucially important for patients with DoC. Indeed, ”efforts to detect covert consciousness have been identified as a moral imperative in light of the consequential impact that this finding may have on clinical decision making, prognosis, family perceptions and neurorehabilitation” ([1], p. 3297).

### 2.2. Non-, Un-, and Sub-Consciousness in Patients with DoC

Application of the OA methodology to the EEG analysis in patients with DoC when subjective awareness is either weakened (MCS) or lost completely (VS/UWS) revealed, as was expected, a profound alteration in the OA of the brain–mind nested hierarchy. Specifically, it was documented [19] that, when compared to fully conscious subjects, who displayed the presence of normal, rather large, relatively long-lived, and stable neuronal assemblies [136], the state of *un*consciousness in VS/UWS patients was characterized by the small, short-lived, very unstable, and functionally disconnected neuronal assemblies. The patients in MCS were characterized by the intermediate level of neuronal assemblies’ functional organization, following the ratio of NORM > MCS > VS/UWS [19]. These results generally imply that the *un*conscious brain is composed of several tiny, causally independent, and very unstable functional units (neuronal assemblies) that generate rather small, short-lived, and unstable local electromagnetic fields that lack mutual integration. However, could we draw a clear conclusion regarding the subjective state of patients in VS/UWS (and in MCS as well) in light of the distinction between *non*consciousness and *un*consciousness that was discussed above? In fact, the OA methodology for EEG analysis makes this possible.

The vast body of empirical evidence, recapitulated by Velmans [137] and Fingelkurts and Fingelkurts [138], indicates that for phenomenal consciousness to exist, brain states, that are instantiated by the multiple local electromagnetic fields, must last longer than the time it takes to complete the simplest processing or cognitive operations, which is of the order of several hundreds of milliseconds. Up until that point, however, the neuronal assemblies are capable (by means of local electromagnetic fields) of performing high levels of perceptual analysis, meaning extraction, cognitive processing, and action formation, all of which are completely *un*conscious (for further discussion, see [139,140,141]). At the same time, OA analysis of an EEG makes it possible to go further than that and determine the level below which already *non*conscious (nonphenomenal) processes take place (Figure 3). The OA analysis of DoC patients’ EEGs yielded the following findings [19]: the local electromagnetic fields lasting about 300 milliseconds were linked to a reduced expression of phenomenal consciousness (*sub*consciousness) typical of MCS and *un*consciousness typical for some patients in VS/UWS (those who later develop some level of consciousness [142]); whereas a persistent state of *un*consciousness was linked to local electromagnetic fields lasting about 220 milliseconds [142]. However, if the lifespan of electromagnetic fields is extremely short, such as below a few tens of milliseconds, and thus likely lies within the stochastic level, then already only *non*conscious, purely neurophysiological operations are carried out that do not belong to the mental domain whatsoever [19].

Additionally, the elemental processing modules (transient functional neuronal assemblies) of the brain ought to be stable enough (Figure 3), in addition to being long enough, to guarantee that they will not further decompose into smaller or even singular elements (neurons), resulting in the total loss of any integration, the certain metastable level of which is considered crucial for the emergence of phenomenal consciousness [24,25] (see, also [95,143]). In general (Figure 3), neuronal assemblies in most VS/UWS patients were highly unstable, indicating *un*conscious processes; however, in some VS/UWS patients, the level of instability approached a stochastic or random level, indicating processes that were already *non*conscious. [19]. The patients in MCS exhibited an intermediate level of neuronal assemblies’ stability (between VS/UWS and healthy subjects) that is sufficient enough (at least transiently) to keep their functional structure (and related electromagnetic fields) that is capable supporting a *sub*conscious process with fluctuating transient episodes of phenomenal consciousness [19].

Furthermore, as the OA framework suggests [24,25], each individual local electromagnetic field that is generated by a functional transient neuronal assembly only presents a partial aspect of the phenomenal object, memory, sensation, concept, or thought, while the wholeness of the phenomenal percepts, images, or thoughts is brought into existence by joint entanglement of numerous local electromagnetic fields generated by many distributed transient neuronal assemblies in the brain. In this respect, results from the OA analysis of DoC patients’ EEGs showed (Figure 3) that during the VS/UWS state, there was either a very weak or nonexistent coupling between electromagnetic fields generated by neuronal assemblies located in different cortex areas, indicating either an *un*conscious state (in the case of extremely low synchrony) or a *non*conscious state (in the case of non-existent synchrony) [19]. At the same time, during minimally expressed consciousness (patients in MCS), the cortex was capable of maintaining ‘fragile binding’ states, when various neuronal assemblies displayed a transient but sufficiently robust engagement in functional coupling of their electromagnetic fields [19]. We suggest that during such episodes, patients in MCS would have brief periods of phenomenal awareness, which is clinically referred to as ‘fluctuating’ consciousness [144,145], but otherwise they primarily remain in a *sub*conscious state (Figure 3). ”Therefore, we may speculate that any decrease in such dynamic interplay would result in the situation where raw sensory stimuli (coming from both the outside and within the organism) dominate; and in the case of significant decrease, it would result in a situation where raw sensory stimuli could not be ever integrated in the context of a personally meaningful narrative. Under such condition, a person would very much be the victim of his/her environment, just a passive recipient; things would just happen to such a subject all the time exactly as in the VS and to a lesser extent in MCS patients” ([19], p. 125).

Overall, these findings stress ”the importance of assessing residual operational architectures, which may support [some] subjective awareness, in patients with disorders of consciousness, whose consciousness expression can be underestimated using traditional clinical bedside evaluation” ([19], p. 123). While this approach is very helpful in delineating the border-zone states of phenomenal consciousness and separating the *un*conscious from the *non*conscious processes in patients with DoC, the information about personal identity or experiential selfhood of patients is not readily accessible at this level of analysis and description. At the same time, it is precisely because of the experiential selfhood that we are able to comprehend the ethical and moral significance of what makes life worth living [6] (see also [1,2,7]), as only a self-conscious being can have preferences regarding how its life goes, something that gives the being an interest in continuing to live [2]. Hence, clarity on the presence of first-person phenomenology and the sense of agency of selfhood is needed for patients with DoC and their families. It is also critical to recognize the difference between the state of absence of self-consciousness and the state of awareness of selflessness. This can be accomplished by conducting an objective examination of experiential selfhood via EEG screening [146], which may provide a window through which the phenomenology and moral significance of selfhood could be rigorously anchored in empirical research [1,2,6,7].

## 3. Experiential Selfhood

An experiential selfhood is the most fundamental aspect of conscious experience, which is always pre-reflectively present in healthy subjects as the first-person mode of givenness of the stream of consciousness [147,148,149] (for a current debate on this topic, see [150,151]). In words of Sass, ”this most fundamental sense of selfhood involves the experience of self not as an object of awareness but as an unseen point of origin for action, experience, and thought” ([152], p. xii). Likewise, Zahavi offered a more nuanced description: “…the experiential self—is not a separately existing entity—it is not something that exists independently of, in separation from, or in opposition to the stream of consciousness—but neither it is simply reducible to a specific experience or (sub)set of experiences; nor is it, for that matter, a mere social construct that evolves through time. Rather, it is taken to be an integral part of our conscious life. More precisely, the claim is that the (minimal or core) self possesses experiential reality and that it can be identified with the ubiquitous first-person character of the experiential phenomena” ([153], p. 18).

### 3.1. Neurophenomenology of the Experiential Selfhood

The neurophysiological three-dimensional construct model of the complex experiential selfhood was recently developed within the OA framework (for a detailed description and empirical data, see [146]). This triad model of selfhood captures the multifarious diversity of self-awareness [123,154] by accounting for the phenomenological distinctions between three central aspects of selfhood that are commensurate with one another [155,156]: (i) phenomenal first-person agency (referred to as ‘Self’), (ii) embodiment (referred to as ‘Me’), and (iii) reflection/narration (referred to as ‘I’). A holistic sense of selfhood is produced by the interaction of these three phenomenological elements [146,157]. Neurophysiologically, these three facets of selfhood are mapped to three specific OMs in the so-called brain self-referential network (SRN) [146], which is occasionally referred to as the default mode network [157,158,159,160]. Practically, and based on the most recent empirical evidence from psychiatric and neurologic studies, a group of nine EEG operationally synchronized cortical areas is used to estimate the synchrony strength within the three OMs (anterior OM: F3-Fz-F4 EEG locations; left posterior OM: T5-P3-O1 EEG locations; and right posterior OM: T6-P4-O2 EEG locations), each of which is related to a set of specific functions that can be subsumed under the names ‘Self’, ‘Me’, and ‘I’ (for a further detail, see [146]). Lately, a causal relationship between these three OMs of the brain’s SRN and the three phenomenological characteristics of selfhood that are associated with them was demonstrated experimentally [146]. By the same token, it has been shown that changes in the phenomenology of selfhood during various neuropsychopathologies, such as depression [161], post-traumatic stress disorder [162], and brain damage [163], as well as during various altered states of selfhood [164], predictably follow changes in the functional integrity (indexed by the qEEG operational synchrony measure) in the triad of SRN OMs. For example, phenomenal upregulation of the expression of Self, Me, or I resulted in a significant increase in the functional integrity (indexed by the qEEG operational synchrony measure [89,90]) of the corresponding SRN OMs, whereas conversely, downregulation of the phenomenological sense of the Self, Me, or I led to a significant decrease in the functional integrity of the respective SRN OMs [146].

According to the triad model of selfhood, the anterior OM of the SRN is linked to the phenomenal first-person perspective and the phenomenal sense of agency [146], where agency is defined as (i) the ‘sense of ownership’ of self-relevant perceptions, thoughts, and actions [135,165,166] and (ii) the sense of the implicit first-person mode of givenness that undergoes the subjective experience [147,148,149]. It is labeled the ‘witnessing observer’ or in short the ‘Self’ in the narrowest sense [146]—the phenomenal non-conceptual core in the act of knowing itself [167]. Phenomenologically, every time the ‘Self’ component is upregulated, the person experiences an “…increased sense of being an epistemic agent that possesses increased self-concept clarity, established a self-representational kind of knowledge for the body, as well as epistemic self-control, all of which together are sufficient for creating a phenomenological first-person perspective …. Further, …this phenomenology also contributes to a sense that one has the capacity for selective, top-down attentional control, and also knows that it (oneself) possesses this capacity—thus having the phenomenal ownership” ([146], p. 18). In contrast, when the ‘Self’ component is downregulated, the person reports that there is ‘no-one who thinks or observes’, and as a result, the experience appears to be phenomenologically ‘empty’ [146]. In the extreme case scenario, it assumes “…a complete absence of any form of phenomenal Selfhood, even the minimal form of spatial-temporal frame of reference—unextended point capable of epistemic self-control, as well as the absence of intentional content, complete emptiness, a void” ([146], pp. 19–20).

The SRN’s right posterior OM is associated with (i) the experience of self as an entity normally localized within bodily boundaries through a mechanism of interoceptive and exteroceptive sensory processing, (ii) emotional states related to body, and (iii) autobiographical emotional memories [146]. It is labeled ‘representational–emotional agency’ or in short ‘Me’ [146]. The distinguishing characteristic of the ‘Me’ module is that, as opposed to a phenomenal first-person perspective, here only a purely geometrical first-person perspective is present that originates from within the body representation, indicating an egocentric spatiotemporal self-model [167]. Importantly, rather than being just one more (among many) physical objects, the body is viewed in this context as a ‘vehicle’ that enables one to be a self in the world [168,169,170,171,172,173]. The phenomenological sensation of hyperembodiment is strongly correlated with the upregulation of the ‘Me’ component, thus allowing a globalized form of phenomenological self-identification with the body as a whole. This lays the foundation for a basic, minimal, and pre-reflective aspect of self (the ‘material me’), along with associated emotional states [146]. Persons experience disembodiment, bodilessness, or an out-of-body experience when the ‘Me’ component is downregulated. Further, in the downregulated ‘Me’ state, related experiences include a subjective suspension of time and space, as well as a diminished or absent automatic and immediate sense of physical agency, first-order experiential sense of ownership (that it is me who owns the body), body self-location, body image, and body schema (for further discussion, see [146]).

The left posterior OM of the SRN is linked to the subjective experience of thinking about and reflecting on oneself, which includes (i) momentary narrative thoughts and inner speech and (ii) the reinterpretation of self-related episodic and semantic memory events (autobiographical story telling) [146]. It is labeled ‘reflective agency’ or shorty ‘I’ [146]. This narrative self-reflection relies on the capability for natural language [9,73,74] and brings about the subjective sense of continuity of selfhood over time [75,76,77]. The enhanced ‘I’ component is “…associated with activation of autobiographical memories, comprising of episodic and semantic memories that consist of either concrete and specific items/episodes of personal information that are closely related to events situated in the past or semantic personal information such as general knowledge of personal facts but also general (repeated and extended) events” ([146], p. 17). In addition, there is increased self-reflection and thinking about one’s own narrative self, which necessitates a more precise self-concept expressed in a deeper comprehension of one’s own states, traits, and dispositions [146]. These kinds of narratives usually involve a rather high level of cognition. During reduced ‘I’ component, persons have a phenomenological experience that ”the inner commentator is quiet and the contents of experience could freely change and flow without a story”. Furthermore, in such state “…disconnected thoughts just popped-up ‘in and out’ in the absence of any explicit subjective sense of presence, past, or future, thus indicating disruption in narration and self-reflection that together are a prerequisite for the cognitive self that persists across experiences” ([146], p. 18).

”Utilising this empirically-grounded triad model of Selfhood in the assessment of patients with DoC may help shed light on whether and which patients have full or minimal self-awareness, and which (or all) aspects of Selfhood are present, diminished or absent. Further, keeping with a demonstrated causal link between three aspects of Selfhood and three SRN modules (measured by qEEG)…, clinicians (and relatives) may get insight into the phenomenal experience of a given patient. This knowledge may give at least some hints as to whether the patient enjoys the moral status of the kind and degree that is sufficient for personhood (in other words be a subject of a life) or only to support some aspects of phenomenal self-experience, as for example, embodiment (pleasure and pain)“ ([6], p. 4). To have a full moral status, according to Levy [2], is to have an interest in life, conceive oneself as lasting in time, and be capable of having future-oriented desires, thus having a motivation to continue living. One could make a rough guess that in order to have this kind of moral status, at least the ‘Self’ and ‘I’ aspects of selfhood should be present.

### 3.2. Non-, Un-, and Sub-Consciousness of Selfhood in Patients with DoC

In contrast to healthy neurotypical individuals who have a relatively high, stable, and balanced levels of functional integrity within and between the three modules of the brain’s SRN (Self, Me, and I modules) [157], that are required for supporting first-person perspective taking, an experience of agency and ownership, and a sense of temporal continuity [135,146,147,148,149,166], patients in the VS/UWS had the overall SRN functional integrity at the stochastic or very low levels [174]. Taking into account what was discussed in Section 2 above, we argue here that a stochastic level would correspond to a complete loss of selfhood when only *non*conscious purely neurophysiological process takes place, thus signifying the presence of rigid stimulus-response behavior in earlier phylogenetic animal lineages [175]. This level does not, however, preclude the cognitive operations (including the complex ones) from being executed, such as those that mediate the functioning and self-organization of the biological ‘machine’ [75,76], though without any phenomenological content [68,176]. The next (low) level of SRN functional integrity would then correspond to the *un*conscious level of selfhood expression, thus signifying “…a phenomenological state of selfless, bodiless and timeless presence, characterized by a lack of individual first-person perspective and an ‘emptying out’ of all phenomenological contents, including thoughts” ([6], p. 4). This is consistent with previous other studies that found the functional integrity of SRN is nonexistent or lowest in several conditions that are characterized by a lack of phenomenal self-awareness: it is totally absent in brain death [177], virtually undetectable in coma [178], and extremely low or severely disrupted in VS/UWS [179]. On the other hand, when in an MCS state, some degree of SRN functional integrity may already sustain an unstable or ‘flickering’ sense of self that is neither fully integrated nor completely fragmented (the *sub*conscious), which is similar to dreaming [180] or being in an altered state of consciousness [164]. Phenomenologically, the altered states of self that patients with MCS experience include time distortion, thinking acceleration, and a variety of transcendental phenomena, such as the ‘dissolution’ of the body or an ‘out-of-body’ experience [181,182].

As it is stressed recently, ”Given the critical importance of major ethical decisions (i.e., in particular, withdrawal of life-sustaining therapy) that are often made while dealing with patients in DoC, such patients would benefit from the brain assessment aiming to evaluate the level of functional integrity of SRN and its OMs, and thus infer which patients are at least minimally self-aware and which aspects of selfhood dominate, regardless of whether these patients do not exhibit self-reflective abilities on additional behavioral/instrumental tests” ([6], p. 6). In this context, and in keeping with the previously established causal link between the triad SRN OMs and the three aspects of selfhood [146], a number of conditions that would all be associated with a lack of phenomenal sense of selfhood though characterized by different combinations of functional expressions of the Self–Me–I triad components and, therefore, having a nuanced phenomenology, can be suggested (Figure 4).

(i)The presence of normal or increased functional integrity of the Self-module of the brain SRN with simultaneous marked loss in the functional integrity (disintegration) of both Me- and I-modules (Figure 4): In such a state, there will be ”the feeling of being a phenomenal spatio-temporal (and often extensionless) point, that observes and witnesses itself and the world” ([164], p. 264) brought about by the Self module, which will be co-present with a complete loss of all contents stemming from the sense of disembodiment (that is accompanied by loss of the automatic and immediate sense of physical agency, along with a decrease in the first-order experiential sense of ownership and emotionality [113,124,147,183,184]) linked to a disintegrated Me-module and a lack of thinking, self-reflection, and personal narrative [151,185,186,187,188,189,190] associated with the disintegrated I-module. Additionally, given that it has been demonstrated that the phenomenal sense of time emerges as a result of the embodiment sense sustained across time [115,191,192,193], one should anticipate “a profound alteration in time perception (feeling of timelessness)” ([164], p. 265) when the sense of body disappears. We can define this state as a ‘*witnessing without content*’. According to Metzinger [51,194] and considering the recent empirical evidence [146,164], such ‘witnessing’ sensation is nevertheless “…sufficient for creating a phenomenological centre of gravity and self-identification that is tied to an individual phenomenological first-personal givenness…” ([164], p. 266), though as a ‘thin’ or ‘nonexplicit’ phenomenal experience (see [195,196]). Thus, referring to Levy’s ‘full moral status’ postulate [7], a patient in this state would have personhood with a distinct individual first-person perspective, though there would be a loss of awareness that it is the same person temporally extended across the time. This is because, for that, the intact self-narration and autobiographical memory that are instantiated by the I-module should be present, but they are not due to its (I-module) functional disintegration. Indeed, having access to autobiographical knowledge is essential for a cognitive selfhood because what one did and experienced in the past defines one’s personal identity in the present and actually shapes how one imagines the self in the future [197]. In this respect, given Levy’s definition [7], we may conclude that this state only ensures a partial moral status with a lack of experience of ‘life worth living’ [7,8,198].(ii)A significant loss of the Self-module’s functional integrity (disintegration) despite the normal levels of the Me- and I-modules’ functional integrity (Figure 4): Such a combination in the OM triad’s functional integrity, when viewed in light of the previous study’s findings regarding the causal relationships between the functional integrity of the three SRN OMs and their corresponding three phenomenological aspects of selfhood [146], could indicate that in this state there is phenomenal ‘emptiness’ or ‘nothingness’ because there is no one to whom the experience is happening, not even the unextended point capable of epistemic self-identification [51,194]. Since the other two brain SRN modules (Me-module and I-module) are functioning normally, there will be phenomenal states related to stimuli originating from both the outside and within the organism and that are also stored as memory traces, but they will not be integrated within the first-person meaningful perspective [146]. Reframing Baars et al. [199] in such a state, there is no blockage of the phenomenal objects of consciousness; rather, *the observing subject is not at home*. Furthermore, concerning Levy’s ‘full moral status’ postulate [7], it is reasonable to expect that the patient will not have a full moral status while being in this state because, despite the fact that autobiographical memory events are phenomenally present, they are not present to anyone since there is no witnessing agent who would be able to observe them from the phenomenal first-person perspective and to whom the experiences are occurring [146,164].(iii)A profound loss of the functional integrity (total disintegration) of all three brain SRN models (Self, Me, and I) (Figure 4). Such a state would signify the complete absence of all self-relevant phenomenological content characterized by the ”selfless, objectless and timeless presence” ([164], p. 272), when the self-referential mechanisms of forming the phenomenological events are suspended [51]. This state is generally characterized by a marked lack of individual first-person perspective, sense of witnessing agency, and ownership [146,164]. Additionally, subjective time (a sense of presence, past, or future) does not present anymore [146,164]. We define this state as a ‘*complete dissolution of experiential selfhood*’. This state could not sustain any phenomenality related to selfhood, and, thus, there is no sense in considering any moral status [7] for patients who are in such a state of lack of ”locus of experience and self-ascription” ([146], p. 23).

**Figure 4 brainsci-13-00814-f004:**
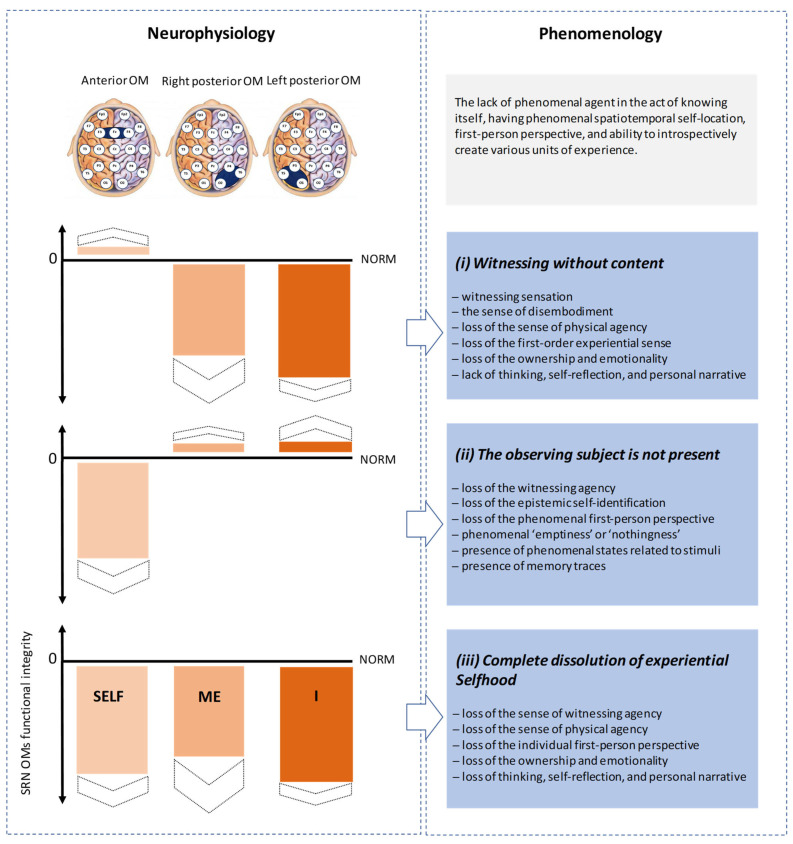
A schematic illustration of the correspondence between the neurophysiology of Self–Me–I OMs and the lack of phenomenal sense of selfhood. OMs: operational modules of the brain self-referential network; SRN: brain self-referential network. The horizontal axis represents OMs’ functional integrity in healthy fully self-conscious subjects (NORM), and it is taken as a ‘0’ for every given OM. The doted arrows depict a putative variability of the functional integrity within each individual OM. The functional disintegration should be rather dramatic in order to reach the loss of self-consciousness. The schematic brain’s cortex maps above the graphs indicate the positions of three OMs (dark blue shapes). Explanations are in the text.

Thus, despite the fact that all three conditions are blatantly tied to a loss of sense of selfhood, one may argue that these three conditions, distinguished by the three different combinations of brain SRN Self–Me–I components, have unique and nuanced phenomenological profiles and, thus, are far from being the unequivocal phenomenon (Figure 4). Indeed, the lack of experiential selfhood “…can take different forms where various aspects or components of Selfhood are affected or expressed differently” ([164], p. 274) (for a similar inference, see [123,146,156]). It is important also keep in mind that the entire picture is further complicated by the fact that “…every studied component of Selfhood comprises several low-level components. For example, the Me-component subsumes body image, body perception, body orientation, ownership, geometrical first-person perspective and physical agency; the I-component includes reflection, rumination, narration, autobiography, thoughts’ structure and speed; the Self-component comprises phenomenal centre, phenomenal first-person perspective, epistemic certitude, witnessing observer” ([164], p. 275).

Additionally, the three conditions presented above are extreme cases of the functional disintegration of one, two, or three modules of the Self–Me–I triad. Different degrees of functional breakdown would, however, likely be present in actual clinical practice (Figure 4). This suggests the necessity for more research to pinpoint functional disintegration thresholds related to the phenomenology described above.

Furthermore, in the relation to the full moral status [7] that provides the phenomenological experience of ‘life worth living’ when the person has an interest in life and has desires and motivation to live [2], it is reasonable to expect that at least two modules of the brain’s SRN that are responsible for the ‘Self’- and ‘I’ components of selfhood should be functionally intact [6]. This will ensure the presence of the phenomenal agent in the act of knowing itself, having phenomenal spatiotemporal self-location, first-person perspective, and ability to introspectively create various units of experience, thus leading to the subjective inner life, which includes autobiographical, narrative, and social (including future-oriented) selves that matter morally [2,6,146,164].

## 4. Conclusions

To summarize, we proposed in this opinion paper to adopt an approach that avoids generalizing across distinct states and conditions that are lacking consciousness using the same term ‘unconsciousness’ under the presumption that they all share the same underlying property of (un)consciousness. Further, we suggested the conceptual decoupling of at least three well-defined states (*non*conscious, *un*conscious, and *sub*conscious) that allow a more nuanced characterization of what is commonly referred to as ‘unconsciousness’, and thereby relate the diverse neurophysiological mechanisms to distinct ‘flavors’ of phenomenological experience in these states. This is especially important in the borderline states of consciousness often present in the patients with DoC, where the personal identity or selfhood is diminished or lost completely. All these issues are central “…in decisions to limit or continue life-sustaining treatment, speaking powerfully to the centrality of consciousness to the concept of personhood and to what makes life worth living” ([1], p. 3292) (for a further discussion, see [22,200]). The ethical significance of such decisions guided by the moral value of these states highlighted by the advanced neurotechnologies becomes immediately apparent when viewed in this manner. This is also true in the case of arguments for ‘higher brain death’ criteria, in which the legal definition of ‘death’ is based on the person’s lack of consciousness rather than the absence of all brain activity [201].

## Figures and Tables

**Figure 2 brainsci-13-00814-f002:**
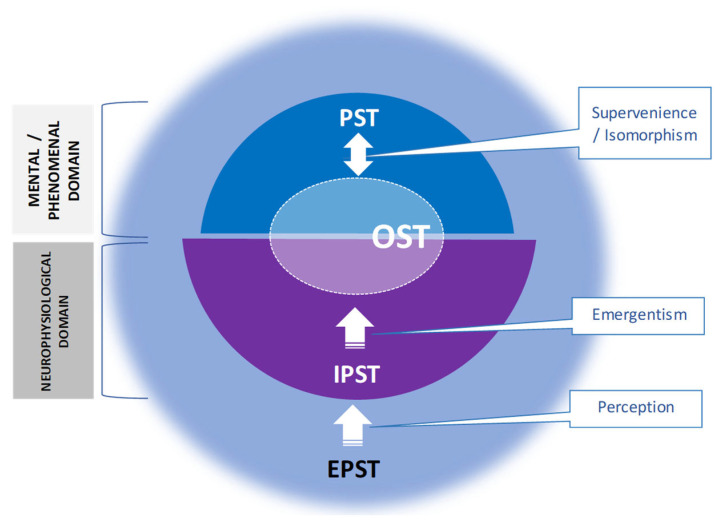
Operational architectonics (OA) of brain and mind functioning. The spatiotemporal patterns of the outside physical environment constitute external physical space–time (EPST). Highly organized and dynamic spatiotemporal patterns of the nested extracellular electromagnetic fields generated by local transient and functional neuronal assemblies constitute the internal physical space–time of the brain (IPST). Operational spatiotemporal pattern (OST) is the abstract (virtual) space and time that are ‘self-constructed’ in the brain each time a particular mental operation needs to be performed. OST level is an emergent property of the brain itself (IPST level). Self-presenting phenomenal spatiotemporal patterns at the higher level of abstractness constitute phenomenal space–time (PST). OST is isomorphic to PST. The phenomenal level (PST) supervenes on the operational level of brain organization (OST). In this model, the OST level (the nested hierarchy of the electromagnetic brain fields) represents the constitutive mechanism of phenomenal consciousness; it ties the phenomenal/subjective (PST) and neurophysiological/physical (IPST) levels together through the shared notion of operation. The PST, as a whole, in turn serves as a transparent surrogate of the EPST of the world. (For further detailed description and discussion, see [24,25,93,94]).

**Figure 3 brainsci-13-00814-f003:**
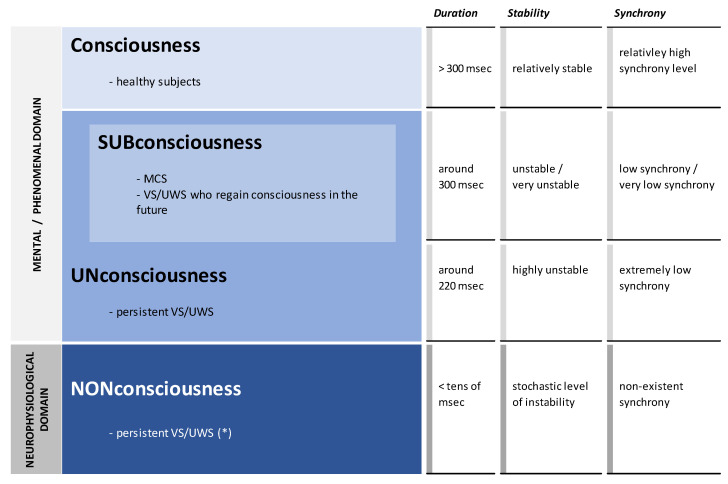
Sub-, un-, and non-consciousness in patients with DoC and brain OA. DoC: disorders of consciousness; OA: operational architectonics of brain and mind functioning; MCS: minimally conscious state; VS: vegetative state: UWS: unresponsive wakefulness state; Duration: lifespan (duration) of local electromagnetic fields; Stability: the level of stability of local electromagnetic fields; Synchrony: the level of mutual coupling (synchrony) among multiple local electromagnetic fields; Msec: milliseconds. Explanations are in the text. The asterisk (*) designates the fraction of DoC patients who should have a suitable categorization in the future in order to be separated from the comparable patients in VS/UWS, who are likewise ‘persistent’ but already possess the phenomenal (though unconscious) states.

## Data Availability

Not applicable.

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
