# Peer review of "Patients with Disorders of Consciousness: Are They Nonconscious, Unconscious, or Subconscious? Expanding the Discussion"

_brainsci, 2023, doi:10.3390/brainsci13050814_

Round 1

Reviewer 1 Report

The authors of the article titled "Patients with Disorders of Consciousness: Are They Nonconscious, Unconscious, or Subconscious? Expanding the Discussion" researched/offered an opinion about an approach that avoids generalizing across distinct states and conditions that are lacking consciousness using the same term 'unconsciousness' under the presumption that they all share the same underlying property of (un)consciousness (unconsciousness, nonconsciousness, and subconsciousness). Also, they provided a brief overview of the state of the field of unconsciousness and showed how a rapidly evolving electroencephalogram (EEG) neuroimaging technique might offer tools (empirical, theoretical, and practical) to approach unconsciousness and improve our ability to distinguish consciousness from unconsciousness and also nonconsciousness with greater precision, particularly in borderline cases (as is typical in patients with Disorders of Consciousness (DoC)). Furthermore, they also described three distant notions of (un)consciousness (unconsciousness, nonconsciousness, and subconsciousness) and discussed how they relate to experiential Selfhood, which is essential for comprehending the moral significance of what makes life worth living.

This is an exciting research paper. The manuscript consists of four chapters: 1. Introduction, 2. (Un)Consciousness, 3. Experimental Selfhood, and 4. Conclusion. In my opinion, the manuscript is well structured, well narrated, and strongly supported by literature references. Every chapter is thoroughly covered. I have no objections or concerns.

Well done.

Author Response

The Reviewer accepts the reliability and the importance of the article, stating that “the manuscript is well structured, well narrated, and strongly supported by literature references. Every chapter is thoroughly covered. I have no objections or concerns”.

The reviewer does not have any comments or suggestions. 

Reviewer 2 Report

We read with interest the opinion article titled “Patients with Disorders of Consciousness: Are They Nonconscious, Unconscious, or Subconscious? Expanding the Discussion” by Fingelkurts and Fingelkurts where the authors raise ethical questions about how to recognize  autonomy in disorders of consciousness (DoC) patients. The authors discuss and argue about the notion of three concepts of consciousness, unconsciousness, and   nonconsciousness with the help of  the electroencephalogram (EEG) neuroimaging technique

Comments:

This is a very well-written piece of literature that combines both science and philosophy to dissect the three terms of consciousness involving unconsciousness, subconsciousness, and nonconsciousness. The work is complex and the authors try to offer a road map to distinguish these three entities which have implications on decisions related to life-sustaining treatment.

One thing is that the preset piece would be a bit complex to the average neuroscientist reader/researcher and I would suggest that more simplified schematics be added.

Along the same line of neuroscience, it would be beneficial if the authors could dwell on the neural mechanism of the regions and neural circuits involved in these concepts and how they are regulated.

Author Response

The Reviewer accepts the reliability and the importance of the subject of the article, stressing that “This is a very well-written piece of literature that combines both science and philosophy to dissect the three terms of consciousness involving unconsciousness, subconsciousness, and nonconsciousness. The work is complex and the authors try to offer a road map to distinguish these three entities which have implications on decisions related to life-sustaining treatment.”. The Reviewer has two minor comments/suggestions.

1. In the first comment, the Reviewer expressed a thought that the text would be a bit complex to the average neuroscientist reader/researcher and the Reviewer would suggest that more simplified schematics could be added.

Reply: It is not clear to what part of the text the Reviewer is referring. The (un)consciousness part is presented in a rather simple way in the Figure1. We thought that it could be that the Reviewer is referring to the part of the text that deals with the lack of Selfhood; - so we introduced in the revised manuscript a new Figure 4 that schematically illustrates the neurophenomenology of three distinct states of lack of self-consciousness.

2. In the second comment, the Reviewer mentioned that it would be beneficial if the authors could dwell on the neural mechanism of the regions and neural circuits involved in these concepts.

Reply: In fact, all those concepts do not relay on any specific brain region or neural circuit. We make it clear in several parts of the text in the revised manuscript: line 157-159: “It is noteworthy that this level encompassing all physiological processes in entirety and not restricted to any particular localized neural circuit or brain region.”; line 330-331: “Again, this level is not restricted to any particular localized neural circuit or brain region, – this is the functional property of the whole brain.”; line 347-350: “The complexity of phenomenality is linked to the complexity of the electromagnetic fields generated by a set of transient functional neuronal assemblies which are located in different regions of the brain. The exact set varies contingent on the involved qualitative phenomenal features they present (for further discussion, see [24,25]).”; and also line 364: “for a relation to neurophysiology and concrete brain areas/circuits, see [129].”

Reviewer 3 Report

This opinion paper provides a brief overview of the field of unconsciousness and demonstrates how a rapidly evolving electroencephalogram (EEG) neuroimaging technique may offer empirical, theoretical, and practical tools to approach unconsciousness and improve our ability to distinguish consciousness from unconsciousness and also nonconsciousness with greater precision, particularly in borderline cases. In addition, three distinct concepts of (un)consciousness are presented (unconsciousness, nonconsciousness, and subconsciousness).

This opinion paper addresses an important and challenging topic by providing adequately reasoned arguments. Figures are useful. The concluding statements summarize the important claims and present the readers with the bigger picture.

I would like to make the following minor suggestions:

-In the introduction, the objective of this article is mentioned. However, it would be useful to provide an additional brief description of the structure of the paper.

-In the second section, we suggest the research of Drigas and Mitsea about subconsciousness techniques.

-There may be a few grammatical, punctuation, and/or spelling errors, but overall, they do not detract too much from reading the paper. It would be useful to make a final check.

Author Response

The Reviewer accepts the reliability and the importance of the subject of the article, stressing that “This opinion paper addresses an important and challenging topic by providing adequately reasoned arguments. Figures are useful. The concluding statements summarize the important claims and present the readers with the bigger picture”. The Reviewer has three minor comments and recommends.

1. In the first comment, the Reviewer mentioned that it would be useful to provide a brief description of the structure of the paper.

Reply: Done. We have rephrased the paragraph describing the aim of this opinion paper (line 78-84) in order to incorporate the information about the structure of the paper as: “In this opinion paper, we will provide a snapshot of the current state of affairs in the field of unconsciousness (Section 2) and demonstrate how a rapidly developing electro-encephalogram (EEG) neuroimaging technique (Section 2.1) may provide empirical, theoretical, and practical tools to approach the unconsciousness and to improve our ability to distinguish consciousness from unconsciousness with greater precision, particularly in cases that are borderline (Section 2.2) or involve diminished experiential Selfhood (Section 3).

2. In the second comment, the Reviewer proposed to mention in the Section 2 the work of Drigas and Mitsea about subconsciousness.

Reply: We would like to thank the Reviewer for recommending a very relevant work to include in our analysis. Done. It is the reference 61 in the revised manuscript.

3. In the third remark the Reviewer mentioned that even though he/she is not an expert in English, he/she thinks that there might be a few punctuation errors and/or typos.

Reply: We have re-checked the whole text for potential mistakes and correct them.